# Discovery of the New Leaf Rust Resistance Gene *Lr82* in Wheat: Molecular Mapping and Marker Development

**DOI:** 10.3390/genes13060964

**Published:** 2022-05-27

**Authors:** Harbans S. Bariana, Prashanth Babu, Kerrie L. Forrest, Robert F. Park, Urmil K. Bansal

**Affiliations:** 1Plant Breeding Institute, School of Life and Environmental Sciences, Faculty of Science, The University of Sydney, 107 Cobbitty Road, Cobbitty, NSW 2570, Australia; prashanthbabuh@gmail.com (P.B.); robert.park@sydney.edu.au (R.F.P.); urmil.bansal@sydney.edu.au (U.K.B.); 2Division of Genetics, Indian Agricultural Research Institute, New Delhi 110012, India; 3AgriBio, Centre for AgriBioscience, Agriculture Victoria, 5 Ring Road, Bundoora, VIC 3083, Australia; kerrie.forrest@agriculture.vic.gov.au

**Keywords:** leaf rust, gene mapping, marker development, wheat, triple rust resistance

## Abstract

Breeding for leaf rust resistance has been successful worldwide and is underpinned by the discovery and characterisation of genetically diverse sources of resistance. An English scientist, Arthur Watkins, collected pre-Green Revolution wheat genotypes from 33 locations worldwide in the early part of the 20th Century and this collection is now referred to as the ‘Watkins Collection’. A common wheat genotype, Aus27352 from Yugoslavia, showed resistance to currently predominating Australian pathotypes of the wheat leaf rust pathogen. We crossed Aus27352 with a leaf rust susceptible wheat selection Avocet S and a recombinant inbred line (RIL) F_6_ population of 200 lines was generated. Initial screening at F_3_ generation showed monogenic segregation for seedling response to leaf rust in Aus27352. These results were confirmed by screening the Aus27352/Avocet S RIL population. The underlying locus was temporarily named *LrAW2*. Bulked segregant analysis using the 90K Infinium SNP array located *LrAW2* in the long arm of chromosome 2B. Tests with molecular markers linked to two leaf rust resistance genes, *Lr50* and *Lr58*, previously located in chromosome 2B, indicated the uniqueness of *LrAW2* and it was formally designated *Lr82*. Kompetitive allele-specific polymerase chain reaction assays were developed for *Lr82*-linked SNPs. *KASP_22131* mapped 0.8 cM proximal to *Lr82* and *KASP_11333* was placed 1.2 cM distal to this locus. *KASP_22131* showed 91% polymorphism among a set of 89 Australian wheat cultivars. We recommend the use of *KASP_22131* for marker assisted pyramiding of *Lr82* in breeding programs following polymorphism check on parents.

## 1. Introduction

Leaf rust, caused by *Puccinia triticina* Eriks. (Pt), is a major disease of wheat (*Triticum aestivum* L.) in many parts of the world. A recent global survey of the impact of pests and pathogens in wheat rated leaf rust as the most damaging globally [1]. Deployment of host resistance has been and continues to be an effective measure to control rust diseases, including leaf rust [2,3]. Sources of resistance that condition near-complete protection against avirulent pathogen isolates throughout the entire life of the plant are referred to as all stage resistance (ASR), whereas those that provide low to moderate levels of resistance at the post-seedling stages are classified as adult plant resistance (APR). A combination of more than two APR genes is needed to condition commercially acceptable level of resistance [4,5]. Combinations of ASR and APR genes are desirable to achieve durable control of rust pathogens. In the 1980s, two groups of wheat cultivars became popular in north-eastern Australia that were derivatives of cultivars Hartog (Pavon S) and Cook. Hartog carries the resistance gene combination *Lr1* and *Lr13* and in addition the APR gene *Lr46*. In contrast, Cook carries ASR gene *Lr3a* and APR gene *Lr34*. Leaf rust resistance gene *Lr24* was backcrossed into both backgrounds. *Lr24* and *Sr24* are located on an alien segment and inherit together [6]. Wheat cultivars possessing *Lr24* covered approximately 28% and 20% of 1999 and 2000 crop season receivals by the Australian Wheat Board in eastern Australia, respectively [7]. Similarly, backcross derivatives of cultivars Cook and Hartog with leaf rust resistance gene *Lr37* covered 33.2% and 21.2% receivals in Queensland and New South Wales, respectively, during the 1999 crop season [7].

Despite evolution among Pt populations, the leaf rust resistance gene combination *Lr1* and *Lr13* remained effective for a long time in Australia [8]. The introduction of Pt pathotype 104-(2),3,(6),(7),11 in 1984 was not significantly different from previously present pathotypes in terms of pathogenicity on Australian wheat cultivars [9]. It did however evolve to render *Lr24* [10] and *Lr37* [11] ineffective. A putative somatic hybrid pathotype that carried full virulence for *Lr1* and partial virulence for *Lr13* rendered several hybrid wheats that were heterozygous for these resistance genes susceptible [12]. The Pt pathotype typed as 10-1,3,9,10,11,12 in 2005 was also virulent on genotypes with the *Lr1* and *Lr13* combination and it led to the discovery of a widely ineffective leaf rust resistance gene *Lr73* [13]. Although this pathotype affected some winter wheats, it did not have much effect on spring wheats (H.S. Bariana personal observation). An exotic incursion first detected in 2006, pathotype 76-3,5,7,9,10+Lr37, underwent mutation to give rise to new pathotypes, of which the most significant was one combining virulence for *Lr13* and *Lr24* and was first detected in 2013 (R.F. Park personal observation). Another exotic pathotype 104-1,3,4,6,7,8,10,12+Lr37 was detected in 2014 and it carried virulence for the gene combination *Lr13*, *Lr27*+*Lr31* and *Lr37* (https://www.sydney.edu.au/content/dam/corporate/documents/sydney-institute-of-agriculture/research/plant-breeding-and-production/cereal_rust_report_2016_14_6.pdf (accessed on 20 May 2022). Another significant pathotype 104-1,3,5,7,9,10,12+Lr37 was first detected in 2016 and in the years since has been the most commonly isolated pathotype of Pt in Australia (R.F. Park personal observation).

Eighty-one leaf rust resistance loci have been formally named [14]. Most of these genes belong to the ASR category and follow the gene-for-gene hypothesis [15,16]. Virulence shifts in Pt populations have reduced the effectiveness of several leaf rust resistance genes [17,18], and in some cases has allowed the recycling of defeated genes due to their resistance to newly evolved/exotic pathotypes and the concurrent decline of older pre-existing pathotypes. For example, *Lr23* was ineffective to dominant pre-1984 Pt pathotypes in Australia, and the 1984 exotic introduction 104-(2),3,(6),(7),11 and its derivatives carry partial virulence for this gene (https://www.sydney.edu.au/content/dam/corporate/documents/sydney-institute-of-agriculture/research/plant-breeding-and-production/cereal_rust_report_2012_vol_10_3.pdf (accessed on 20 May 2022)). Virulence for *Lr23* has been rare or absent in Australia since 2004 (R.F. Park personal observation). These pathotypic changes stress the need for continuous discovery and characterisation of APR and ASR loci from diverse germplasm (landraces and wheat wild relatives) collections for sustained wheat production [19].

A set of 838 pre-Green Revolution common wheat landrace genotypes, referred to as the ‘Watkins Collection’ [20], was available in Australia at the Australian Winter Cereal Collection, Tamworth, New South Wales (now Australian Grains Genebank, Horsham, VIC, Australia). These genotypes were screened for leaf rust response at the adult plant growth stages in artificially inoculated leaf rust field nurseries and at seedling growth stages under greenhouse conditions. Entry Aus27352, originally collected from Yugoslavia, showed resistance to the currently predominant Pt pathotypes including 104-1,3,4,6,7,8,10,12+Lr37. This investigation covers mode of inheritance, molecular mapping and identification of markers closely linked with ASR to leaf rust in Aus27352.

## 2. Materials and Methods

### 2.1. Development of a Mapping Population

Aus27352 was crossed with the wheat selection Avocet S (AvS) and an F_2:6_ recombinant inbred line (RIL) population was developed using the single head descent method. Briefly, a single head was harvested from each plant from the F_2_ generation onwards and two seeds from each family were planted in the F_3_, F_4_ and F_5_ generations and a single head was harvested from each family. The whole plant was harvested at the F_6_ generation to generate 200 F_2:6_ RILs.

### 2.2. Seedling Tests

Six sets of parental lines Aus27352 and AvS (8–10 seeds) were sown following the procedure described by Qureshi et al. [21]. Ten to twelve day-old seedlings were inoculated with Pt pathotypes 104-2,3,6,(7) (Plant Breeding Institute culture no. 231), 10-1,2,3,4 (348), 104-1,(2),3,(6),(7),11,13 (547), 10-1,3,9,10,11,12 (592); 104-1,3,4,6,7,8,10,12+Lr37 (634); and 76-3,5,7,9,10,12,13 +Lr37 (625) and incubated for 24 h in a room with 100% humidity before moving to temperature and irrigation-controlled microclimate rooms set at 25 °C. Twenty seeds of each Aus27352/AvS F_3_ line were sown as a single line per pot using the potting mixture outlined in Qureshi et al. [21] and the Aus27352/AvS RIL population was sown as four lines per pot, 10 seeds per line and inoculated with Pt pathotype 104-1,3,4,6,7,8,10,12+Lr37. Variation in leaf rust responses was scored following a scale described in McIntosh et al. [6].

### 2.3. Molecular Mapping

Genomic DNA was extracted from parents and the entire RIL population using a modified CTAB method [22]. DNA was quantified with the Nanodrop 1000 (Thermofisher Technologies, Inc., Waltham, MA, USA) and quality of DNA was tested by agarose gel electrophoresis. Equal amounts of DNA were pooled from 40 homozygous resistant and 40 homozygous susceptible RILs to prepare resistant and susceptible DNA bulks. Both DNA bulks, parental lines and an artificial F_1_ (DNA from 40 random RILS were mixed in equal quantity) were subjected to genotyping using 90K Infinium SNP array at AgriBio, La Trobe University, Bundoora, VIC, Australia to conduct bulked segregant analysis (BSA). Linked SNPs were converted to Kompetitive Allele Specific PCR (KASP) markers using the software Polymarker (http://www.polymarker.info (accessed on 20 May 2022)). These KASP markers were tested on the entire RIL population following a procedure described by Nsabiyera et al. [23].

### 2.4. Statistical Analysis

Chi-squared (χ^2^) analyses were performed to check the goodness-of-fit of the observed leaf rust response and marker loci segregation to the expected genetic ratios. Recombination fractions were computed using MapDisto [24]. A genetic linkage map was drawn using MapChart software version 2.3 [25] to show the graphical representation of locus order.

## 3. Results

Aus27352 and AvS were tested with six Pt pathotypes at the seedling stage and results are presented in Table 1. Aus27352 was susceptible to four pathotypes, and resistant to pathotypes 104-1,3,4,6,7,8,10,12+Lr37 (IT “;11+c”) and 76-3,5,7,9,10,12,13+Lr37 (IT “11-”), whereas AvS produced ITs “3+” against pathotypes 104-1,3,4,6,7,8,10,12+Lr37 and 76-3,5,7,9,10,12,13+Lr37. The low infection type (IT “23-”) produced by AvS against pathotypes 104-2,3,6,(7); 10-1,2,3,4 and 104-1,(2),3,(6),(7),11,13 was conditioned by leaf rust resistance gene *Lr13* and IT “;” against 10-1,3,9,10,11,12 was due to the presence of *Lr73*. Aus27352/AvS F_1_ seedlings produced susceptible infection types when inoculated with pathotype 104-1,3,4,6,7,8,10,12+Lr37 indicating a recessive mode of inheritance. Due to a smaller quantity of seed for 30 Aus27352/AvS-derived F_3_ lines, only 170 lines were tested and monogenic segregation (36 homozygous resistant: 85 segregating: 49: homozygous susceptible, χ^2^_(1:2:1)_ = 1.99; non-significant at *p* = 0.05 and 2 d.f.) for leaf rust resistance was observed. Resistant and susceptible seedlings were counted among 85 segregating families and a recessive mode of inheritance was confirmed (361 resistant: 1152 susceptible, χ^2^_(1:3)_ = 1.08; non-significant at *p* = 0.05 and 1 d.f.). The Aus27352/AvS RIL population was tested with pathotype 104-1,3,4,6,7,8,10,12+Lr37 and segregation at a single locus (homozygous resistant 88: homozygous susceptible 112, χ^2^_(1:1)_ = 2.88; non-significant at *p* = 0.05 and 1 d.f.) was confirmed (Table 2). The underlying resistance locus was temporarily named *LrAW2*.

### 3.1. Molecular Mapping of LrAW2

BSA with the 90 K Infinium SNP wheat array was performed. Thirty-nine SNPs from the long arm of chromosome 2B differentiated resistant and susceptible bulks. These SNPs spanned the 761,279,407 to 794,886,621 bp region of the Chinese Spring physical map (IWGSC RefSeq_V2.0). KASP marker assays were designed for these SNPs and were tested on parental lines. Fourteen SNPs that produced clear clusters (Table 3) were genotyped on the entire RIL population. *LrAW2* was flanked by *KASP_22131* (0.8 cM) on the proximal side (towards centromere) and *KASP_11333* (1.2 cM) on the distal side in the long arm of chromosome 2B (Figure 1).

### 3.2. Testing of Flanking Markers for Polymorphism on a Wheat Panel

Markers *KASP_22131* and *KASP_11333* were assayed on a set of 89 Australian wheat cultivars to assess their roles in marker assisted selection of *LrAW2* in wheat breeding programs. *KASP_22131* and *KASP_11333* amplified G:G and A:A alleles, respectively, in the resistant parental stock Aus27352 and the alternate alleles in the susceptible parent AvS (Table 4). *KASP_22131* was polymorphic in 81 cultivars and produced the AvS allele (A:A), whereas *KASP_11333* was polymorphic in 64 cultivars with the AvS allele (G:G). The amplification of *LrAW2*-linked allele of the closer marker *KASP_22131* occurred in nine cultivars (Correll, Espada, LRPB Kittyhawk, Orion, Chief CL Plus, Gladius, Impose CL Plus, LRPB Arrow and Wedin). Cultivar Correll may carry *LrAW2* or another gene with similar pathogenic specificity based on leaf rust response data against several pathotypes. Kittyhawk lacked this gene and the presence of other effective leaf rust resistance genes in the remaining cultivars did not allow postulation of this locus. Taking into consideration the polymorphism, we recommend that *KASP_22131* can be used for pyramiding of *LrAW2* with other marker-tagged ASR and APR genes for leaf rust resistance.

## 4. Discussion

Leaf rust is prevalent in almost all wheat growing areas globally [2] and causes widespread and at times severe damage to wheat crops [1]. Most of the race specific genes for leaf rust resistance have been overcome by the evolution in local Pt populations and/or by exotic introductions in Australia [8,11,12]. These events however shaped gene-based control measures for leaf rust. The Australian wheat industry relied on the *Lr1* and *Lr13* combination, *Lr24* was introduced in the 1980s following the development of white-seeded recombinants [6]. It was then supplemented by *Lr37* in the 1990s [26]. The value of the APR genes *Lr34* and *Lr46* became widely evident in the 21st century. These observations led to the realisation that durable leaf rust control could be achieved by combinations of ASR and APR genes [3,27,28].

This study identified a new leaf rust resistance locus *LrAW2* in the long arm of chromosome 2B and it is effective against Pt pathotypes 104-1,3,4,6,7,8,10,12 +Lr37 and 76-3,5,7,9,10,12,13 +Lr37, which along with several derivative pathotypes currently prevail in Australia. The markers flanking *LrAW2*, *KASP_21133* and *KASP_11333*, are located in the 788 and 790 Mb regions of the physical map of Chinese Spring, respectively (IWGSC_RefSeq_v2.0). There are two known leaf rust resistance genes located in the long arm of chromosome 2B, *Lr50* [29] and *Lr58* [30]. *Lr50* was introgressed from *T. timopheevi* and *Lr58* from *Aegilops triuncialis.*
*Lr50* (*gwm382*) and *Lr58* (*ncw*-*Lr58*-1) linked markers were tested on the parental lines and 22 RILs from the population. Marker *gwm382* is a dominant marker and produces 139 bp and none of the test lines produced 139 bp amplicon. The *Lr58*-linked marker *ncw*-*Lr58*-1 is a co-dominant marker and produced about 400 bp amplicon in lines carrying *Lr58* and 250 bp in non-carriers. All test lines produced 250 bp products. Both *Lr50* and *Lr58* follow dominant inheritance, whereas *LrAW2* has recessive inheritance. Based on these results *LrAW2* is unlikely to be either of these genes. Hence a permanent gene symbol *Lr82* was allocated to *LrAW2*.

Annotated genes located between the markers flanking *LrAW2* were extracted from the IWGSC genome assembly of wheat cv. Chinese Spring v1.0 using the tool Pretzel (https://plantinformatics.io (accessed on 20 May 2022). Functional annotations for these genes were obtained from the IWGSC RefSeq data repository at INRA (https://urgi.versailles.inra.fr/download/iwgsc/IWGSC_RefSeq_Annotations/v1.0 (accessed on 20 May 2022). Ninety annotated genes (36 high-confidence and 54 low-confidence) were identified between the two KASP markers that flanked *Lr82*. Of these, two genes (TraesCS2B01G608500 and TraesCS2B01G608800) are predicted to encode TIR-NBS-LRR disease resistance proteins, based on the IWGSC RefSeq gene annotation. These could be candidate genes for *Lr82* (Figure 2).

The ‘Watkins Collection’ has been a rich source of new rust resistance genes that are yet to be deployed in agriculture. Previously leaf rust resistance gene *Lr52* was formally named by Canadian workers in a landrace from Iran [31]. Bansal et al. [32] showed close association of *Lr52* with a new stripe rust resistance locus *Yr47*. The *Yr47*/*Lr52* combination is currently being used in Australian and Indian wheat breeding programs (H.S. Bariana personal communication with breeders). Several stripe rust resistance genes have been discovered from this collection [19].

The concept of triple rust resistance is often not addressed holistically, with new cultivars lacking adequate resistance to one or the other of the three rust pathogens [19]. The long arm of chromosome 2B carries several rust resistance genes, two of which that are intriguing from the exotic introduction point of view and can protect Australia against the *Puccinia graminis* f. sp. *tritici* pathotype Ug99 and its derivatives are *Sr28* [33] and *Sr9h* [34]. Although both genes are ineffective to extant pathotypes of *P. graminis* f. sp. *tritici* in Australia, they would assume importance if one or more of the Ug99 group of pathotypes were to be detected in Australia. *Sr9e* could be more useful against predominating Australian pathotypes of *P. graminis* f. sp. *tritici*. In addition, stem rust resistance genes *Sr36* and *Sr39* could be useful candidates for pyramiding [35]. Similarly, stripe rust resistance genes *Yr5a*, *Yr5b*, *Yr43*, *Yr44*, *Yr53* and *Yr72* are effective against a majority of *P*. *striiformis* f. sp. *tritici* pathotypes and are located in the long arm of chromosome 2B [36,37,38,39]. Development of recombinants carrying combinations of *Lr82* with leaf rust, stem rust and stripe rust resistance genes can lead to achievement of durable triple rust resistance.

Markers have been developed for many ASR genes conditioning resistance to leaf rust; *Lr23* [40], *Lr24* [41], *Lr42* [42], *Lr52* [21], *Lr53* [43], *Lr57* [44], *Lr76* [44], *Lr80* [14] and APR genes *Lr34* [45], *Lr48* [23], *Lr49* [46], *Lr67* [47] and *Lr68* [48]. These ASR and APR genes can be pyramided in different combinations to combat evolution in Pt populations. In particular, the combination of *Lr82* with *Lr23* and *Lr24* and at least one APR gene would contribute towards long-lasting control of this disease in Australia.

## 5. Conclusions

This study identified and characterised a new leaf rust resistance gene, *Lr82*, in the common wheat Yugoslavian landrace Aus27352. *Lr82* was mapped 0.8 cM distal to *KASP_22131* in the long arm of chromosome 2B. *KASP_22131* amplified the allele alternate to that linked with *Lr82* in 91% of 89 Australian wheat cultivars. These results support the implementation of *KASP_22131* in marker assisted pyramiding of *Lr82* with other marker tagged rust resistance genes to achieve durable triple rust control in new wheat cultivars.

## Figures and Tables

**Figure 1 genes-13-00964-f001:**
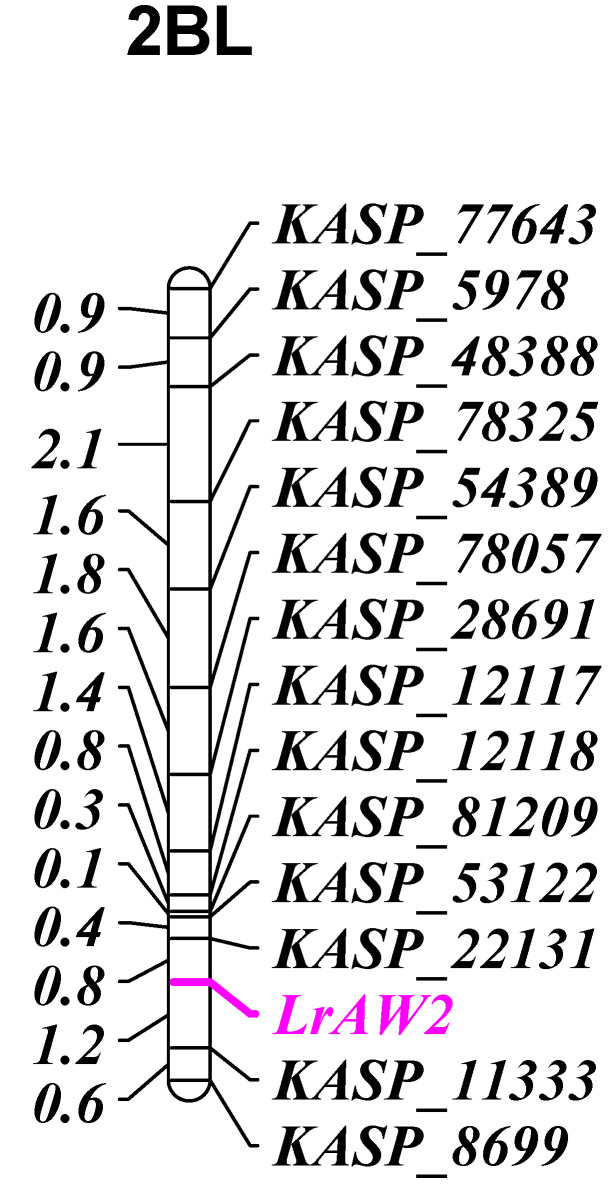
Genetic linkage map of Aus27352/AvS RIL population showing the location of *LrAW2*.

**Figure 2 genes-13-00964-f002:**
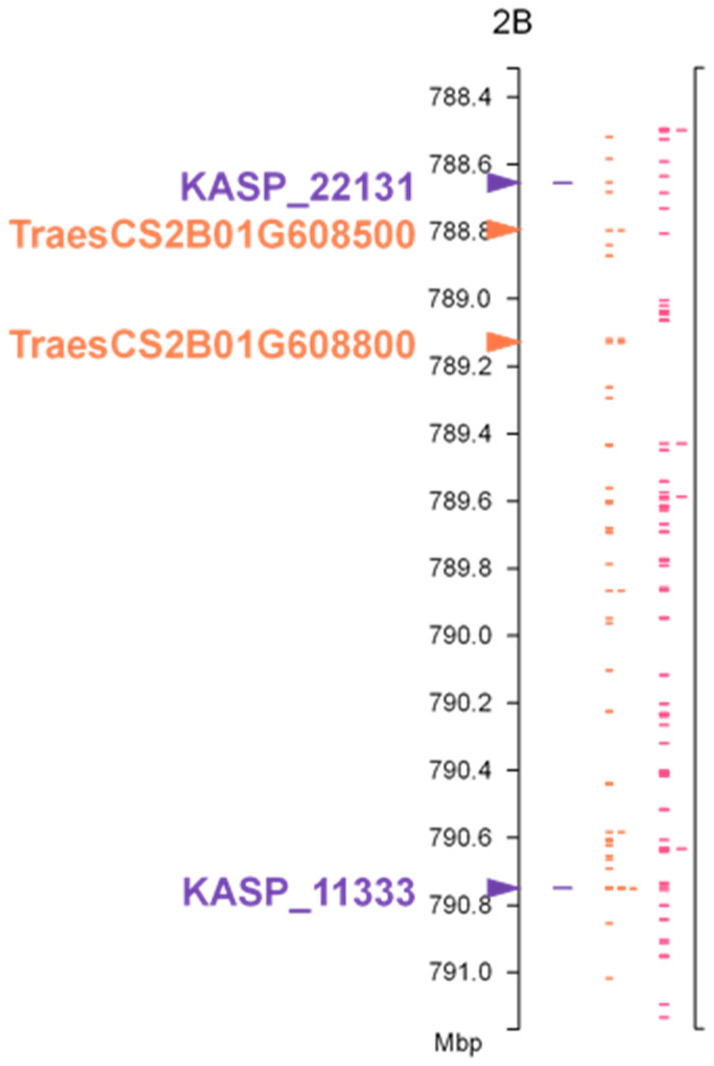
Physical location of annotated genes in the interval containing *Lr82*. Positions (Mbp) shown refer to the International Wheat Genome Sequencing Consortium (IWGSC) Chinese Spring wheat genome assembly v1.0. Orange marks are high confidence (HC) gene annotations and pink marks are low confidence (LC) gene annotations. The location of flanking KASP markers and two genes predicted to encode TIR-NBS-LRR disease resistance proteins are indicated with triangles.

**Table 1 genes-13-00964-t001:** Responses of parental lines to six pathotypes of *P. triticina*.

Pathotype ^	Culture Number *	Aus27352	Avocet S
104-2,3,6,(7)	231	3+	23-
10-1,2,3,4	348	3+	23-
104-1,(2),3,(6),(7),11,13	547	3+	23-
10-1,3,9,10,11,12	592	3+	;
104-1,3,4,6,7,8,10,12+Lr37	634	;11+c	3+
76-3,5,7,9,10,12,13+Lr37	625	11-	3+

^ 1= *Lr20*, 2 = *Lr23*, 3 = *Lr14a*, 4 = *Lr15*, 5 = *Lr3Ka*, 6 = *Lr27*+*Lr31*, 7 = *Lr17a*, 8 = *Lr28*, 9 = *Lr26*, 10 = *Lr13*, 11 = *Lr16*, 12 = *Lr17b*, 13 = *Lr24*; Lr37+ denotes virulence for *Lr37*; the Pt group 104 is virulent on *Lr1* and *Lr3a*; culture 592 is avirulent on *Lr73* and virulent on *Lr1*. Pt group 76 is virulent on *Lr3a* and *Lr13*, and avirulent on *Lr1*. * These numbers represent the unique identity of a purified and confirmed pathotype.

**Table 2 genes-13-00964-t002:** Distribution of Aus27352/AvS RIL population when tested with pathotype 104-1,3,4,6,7,8,10,12+Lr37.

Leaf Rust Response	Infection Type	No. of Lines		χ^2^ _(1:1)_
		Observed	Expected	
Homozygous resistant	;11+	88	100	1.44
Homozygous susceptible	3+	112	100	1.44
Total		200	200	2.88

Table value of χ^2^ (1:1) at *p* = 0.05 is 3.84 and 1 d.f.

**Table 3 genes-13-00964-t003:** KASP markers used to generate a genetic linkage map of Aus27352/AvS RIL population.

Marker	Physical Distance (bp)	Allele 1 ^a^	Allele 2 ^b^	Common
*KASP_77643*	785,138,741	gagccactgatctgatcactt	gagccactgatctgatcactc	tcgtcggtgtttccctgttt
*KASP_5978*	784,551,365	ctgaagcacttcgcccca	ctgaagcacttcgccccg	gaatctacgacgaggctgc
*KASP_48388*	785,890,263	ttgtgtatgtatgttcatttggca	ttgtgtatgtatgttcatttggcg	tctttgtaggttgaaagggct
*KASP_78325*	786,230,653	tgacccatactttgcaacacaa	tgacccatactttgcaacacag	acacgtgatggaaaaggttct
*KASP_54389*	786,105,954	gacatggcggggtcgact	gacatggcggggtcgacc	gaactgacgtgagccatgct
*KASP_78057*	786,229,479	ttacaacgataaggccaccaa	ttacaacgataaggccaccag	cagtgaacttcttcaggcgg
*KASP_28691*	789,608,961	cggatttctggacatcgtca	cggatttctggacatcgtcg	tcaaactttccttgttgttcgtac
*KASP_12117*	788,524,814	tccaccatcccgcagcaa	tccaccatcccgcagcag	aggccttggggacacaatcc
*KASP_12118*	788,524,885	ccccaaggcctctttcgt	ccccaaggcctctttcgg	gccagtttgatgtcgaagagat
*KASP_81209*	788,657,195	tggtagtgctgcaaaacga	tggtagtgctgcaaaacgg	ggtgttggttactacagcagc
*KASP_53122*	788,655,517	gtccaaggccgaggaggat	gtccaaggccgaggaggac	cctgctcagccaacaccattatg
*KASP_22131*	788,656,700	ggctagtgttgtttttgtacca	ggctagtgttgtttttgtaccg	catacaggtagcagatacgcaa
*KASP_11333*	790,751,851	cacggaaccagactggca	cacggaaccagactggcg	gaacccgttctcagcgaat
*KASP_8699*	793,148,680	cagatgatggtggatggtatgtatt	cagatgatggtggatggtatgtatc	atggttgtgggaagcacgaa

^a^ A1 primer labelled with FAM: GAAGGTGACCAAGTTCATGCT; ^b^ A2 primer labelled with HEX: GAAGGTCGGAGTCAACGGATT.

**Table 4 genes-13-00964-t004:** Genotyping of markers flanking *Lr82* for polymorphism on Australian wheat cultivars.

Cultivars *	*KASP_22131*	*KASP_11333*
Aus27352	*G:G*	*A:A*
Avocet S	*A:A*	*G:G*
AGT Katana, Axe, Baxter, Bolac, Carnamah, Catalina, Chara, Cobra, Corack, Crusader, Dart, Derrimut, EGA Bonnie Rock, EGA Burke, EGA Gregory, EGA Wedgetail, EGA Wylie, Elmore CL PLus, Emu Rock, Envoy, Estoc, Forrest, Gauntlet, Gazelle, GBA Sapphire, Giles, Grenade CL Plus, Harper, Impala, Janz, Justica CL Plus, King Rock, Kord CL Plus, Kunjin, Lancer, Lang, Lincoln, Livingston, LRPB Reliant, Mace, Magenta, Mansfield, Merinda, Preston, SF Adagio, SF Scenario, Shield, Sunco, Sunguard, SunMax, Sunvale, Sunzell, Wallup, Westonia, Wyalkatchem, Wylah, Yandanooka, Yitpi, Young	*A:A*	*G:G*
Beaufort, Calingiri, Coolah, DS Faraday, EGA Bounty, Fortune, LRPB Flanker, Mackellar, Merlin, Naparoo, Ninja, Phantom, Scout, Sentinel, Spitfire, SQP Revenue, Strzelecki, Suntop, Trojan, Ventura, Waagan	*A:A*	*A:A*
Correll, Espada, LRPB Kittyhawk, Orion	*G:G*	*A:A*
Chief CL Plus, Gladius, Impose CL Plus, LRPB Arrow, Wedin	*G:G*	*G:G*

* Leaf rust responses of these cultivars can be viewed at https://www.sydney.edu.au/content/dam/corporate/documents/faculty-of-science/research/life-and-environmental-sciences/cereal-rust-research/cereal-rust-report--2020-vol-17-3.pdf (accessed on 20 May 2022).

## Data Availability

All data are given in the manuscript. Publicly available statistical tools are used in this study.

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
