# Peer review of "Discovery of the New Leaf Rust Resistance Gene Lr82 in Wheat: Molecular Mapping and Marker Development"

_genes, 2022, doi:10.3390/genes13060964_

Round 1

Reviewer 1 Report

Authors described the discovery of a common wheat genotype “Aus27352” from Yugoslavia with moderate resistance to certain leaf rust pathotypes. A RIL population was prepared and a genetic locus on chromosome 2BL was mapped as LrAW2. Here are some of my major concerns about this manuscript:

  1. The introduction part of the manuscript has mainly focused on the Pt virulence aspect. More information about genetics and cloning of Lr-ASR genes should be introduced.
  2. Pt pathotypes with only laboratory names provide very few information about their virulent profiles. Standard names, for example PHTT, designated based on phenotypes of Lr isogenic lines are necessary.
  3. How many coding genes between KASP_22131 and KASP_11333 in the reference genome? Are there any genes encoding NBS-LRR type protein?
  4. Phenotypes of 90 Australian wheat cultivars to Pt pathotype 104-1,3,4,6,7,8,10,12+Lr37 should also be provided.
  5. Line 200-202: To distinguish LrAW2 with Lr50 and Lr58, or as in the title to demonstrate “a new leaf rust resistance locus”, isogenic lines carrying Lr50 and Lr58 should be tested against Pt pathotypes of 04-1,3,4,6,7,8,10,12+Lr37 and 76-3,5,7,9,10,12,13+Lr37. Molecular markers for Lr50 and Lr58 should also be used on Aus27352/AvS RIL population or selected progenies.

Minor issues:

Although Aus27352 may belong to “Watkins Collection”, I did not see the necessity of the demonstration in Line 11-13.

Line 28: “Pt” in italic. Also in Line 51, 55…

Explain “Culture Number” in Table 1. Biological replicates?

Line 156: typo SNPs.

Line 237: a space is missed between “KASP_22131” and “in”.

Author Response

Thanks for your critique and here is my responses to your queries:

Reviewer 1

Authors described the discovery of a common wheat genotype “Aus27352” from Yugoslavia with moderate resistance to certain leaf rust pathotypes. A RIL population was prepared and a genetic locus on chromosome 2BL was mapped as LrAW2. Here are some of my major concerns about this manuscript:

  1. The introduction part of the manuscript has mainly focused on the Pt virulence aspect. More information about genetics and cloning of Lr-ASR genes should be introduced.

Author response: The importance of this manuscript lies in identification of a gene that was in-effective against older Pt pathotypes and showed high level of resistance against the newly detected pathotypes. Therefore, pathotypic evolution was chosen to set the scene in introduction and mapping of genes in covered in the discussion section.

  1. Pt pathotypes with only laboratory names provide very few information about their virulent profiles. Standard names, for example PHTT, designated based on phenotypes of Lr isogenic lines are necessary.

Author response: virulence and aviurulence details are added as footnote.

  1. How many coding genes between KASP_22131 and KASP_11333 in the reference genome? Are there any genes encoding NBS-LRR type protein?

Author response: Information is provided in the text and Figure 2. There are two NBS-LRR genes in this region.

  1. Phenotypes of 90 Australian wheat cultivars to Pt pathotype 104-1,3,4,6,7,8,10,12+Lr37 should also be provided.

Author response: These cultivars have been tested several times against the pathotypes used. The information about their response and postulated genes can be viewed at the weblink mention as footnote for Table 4.

  1. Line 200-202: To distinguish LrAW2 with Lr50 and Lr58, or as in the title to demonstrate “a new leaf rust resistance locus”, isogenic lines carrying Lr50 and Lr58 should be tested against Pt pathotypes of 04-1,3,4,6,7,8,10,12+Lr37 and 76-3,5,7,9,10,12,13+Lr37. Molecular markers for Lr50 and Lr58 should also be used on Aus27352/AvS RIL population or selected progenies.

Author response: Markers linked with Lr50 and Lr58 were tested and text is added in discussion.

Minor issues:

Although Aus27352 may belong to “Watkins Collection”, I did not see the necessity of the demonstration in Line 11-13.

Author response: This is mentioned to pay respect to the pioneering work of Arthur Watkins.

Line 28: “Pt” in italic. Also in Line 51, 55…

Author response: I try to keep appreciation simple and without additional formatting. Happy to change if not acceptable.

Explain “Culture Number” in Table 1. Biological replicates?

Author response: Footnote added.

Line 156: typo SNPs.

Line 237: a space is missed between “KASP_22131” and “in”.

Reviewer 2 Report

Identifying new resistance gene  against leaf rust is necessary for wheat breeding and for controlling the rust disease on wheat. In this study, a putative LrAW2 new leaf rust-resistant gene was idetified and located on 2BL. However, the absence of diallele crosses between this gene and other genes that located on 2BL. Therefore, it is not sufficent for supporing this gene is a new gene, but only a putative gene. If this gene is identified to be a new gene. more research works are needed. 

Author Response

Thanks for your comments, my response to queries is given below:

Identifying new resistance gene  against leaf rust is necessary for wheat breeding and for controlling the rust disease on wheat. In this study, a putative LrAW2 new leaf rust-resistant gene was idetified and located on 2BL. However, the absence of diallele crosses between this gene and other genes that located on 2BL. Therefore, it is not sufficent for supporing this gene is a new gene, but only a putative gene. If this gene is identified to be a new gene. more research works are needed.

All tracked changes are accepted, except: 

In Table 4, alleles are corresponding horizontally to different groups of cultivars.

Markers linked with Lr50 and Lr58 were tested on RILs and information added in discussion. This is a unique locus and is named Lr82 by the panel of peers.

Round 2

Reviewer 1 Report

Authors have addressed most of my previous concerns.